# Civic-Moral Education Research in China (1992–2022): A Scoping Review

**DOI:** 10.3390/bs13100819

**Published:** 2023-10-05

**Authors:** Le Shi, Lingjun Chen, Rui Gong

**Affiliations:** School of Education, Shanghai Jiao Tong University, Shanghai 200240, China

**Keywords:** civic-moral education, scoping review, civic morality, civic-moral curriculum, cultural resource, traditional culture, social climate, civic participation

## Abstract

Civic-moral education is a topic that has been widely discussed globally. In China, civic-moral education has a long history and presents different characteristics and methods from other countries due to political, economic, and cultural factors. To summarize the current status of research on civic-moral education in China, we identified 715 papers in 30 years (1992–2022) under related topics and selected a total of 72 papers for further analysis. To show the study trend more clearly, we divided the result section into three parts: the historical and cultural resources of education, the current educational approaches, and the existing challenges in civic-moral education. China is rich in historical resources for civic-moral education, which had an impact across generations. Currently, Chinese educators employ many approaches to improve civic-moral education: building a civic-moral curriculum system, transforming the social climate, and making the most use of the campus. The emphasis on intelligence over morality and the emphasis on knowledge over action, however, may undermine the effectiveness of civic-moral education. To conclude, China attaches great importance to civic-moral education and has come up with many approaches inspired by ancient cultural resources.

## 1. Introduction

Civic morality is a code of conduct closely related to public life that fosters “commitment to a collection of democratic principles and values” [1], such as loving the motherland, obeying the law, upholding social justice, and serving society [2]. The connotation of civic morality is also constantly enriched with time. The new inclusion of environmental protection, resource conservation, wildlife protection, and online etiquette as part of civic morality conduct reflects this trend [3]. Civic morality’s role in maintaining public order, social stability, and harmony is becoming increasingly prominent in China [4,5]. Along with the rapid social change, the development of the internet, and the increased international communications, the scope of the public is growing rapidly. These changes require people to have good civic morality to properly handle various complex issues [6]. 

The importance of civic morality calls for civic-moral education. The realization of civic morality requires both moral education and civic education [7], which is the education of the basic norms and requirements of the citizens in public life [8]. Many scholars discuss civic and moral education together. The most commonly used terms include “civic-moral education”, “the moral dimension of civic education”, and “civic virtue education”, which reflect the undividable moral foundation for civic education [9,10,11,12]. To be consistent, this article uses the term “civic-moral education” (gongde jiaoyu, 公德教育). Chinese scholars revived traditional moral theories tracing back to the Axial Age and realized the importance of civic-morality education [13,14,15,16]. Civic-moral education guides people to care for others and serve society, thus pursuing happiness beyond individuals and obtaining relational satisfaction and self-esteem from social life [17]. Through educating, persuading, and even supervising [18], people can perform under the guidance of civic-moral spirits, as well as social long-term interests [19]; meanwhile, their civic intelligence and civic performance are improved through various practices [6]. According to the official document [20], schools are the priority places for cultivating civic morality in China, and they offer civic science courses for students at different stages. *Ethics and the Rule of Law* is offered to elementary and junior high school students, and *Ideology and Politics* is offered to senior high school students. At the college level, *Ideological and Moral Cultivation and Legal Basis* is a mandatory course.

Over the past 30 years, some reviewing work has been performed regarding civic-moral education in China, but they have certain limitations. Cheng et al. reviewed the literature to introduce moral education from K-12 to higher education in China together with the highlights and limitations in each school stage [21]. However, civic values are not concluded in the scope of the review, and it mainly introduced practices, not research. Only two other Chinese articles reviewed the current state of research on civic-moral education in colleges [22,23], for which the focus was on studies targeting college students’ civic morality levels, issues, and related factors, ignoring studies on other populations.

There are also general review works on relevant fields, but none of the reviews included studies focusing on situations in China. Additionally, they did not include relevant issues of morality education, which are closely tied to civic education. Bennion and Laughlin summarized the papers published by the *Journal of Political Science Education (JPSE)* from 2005 to 2016, which documents the best practices in civic education, presenting some findings on creating engaged citizens, voter education and mobilization activities, stimulations, service-learning, global citizenship, internships, extracurricular programs and activities, and others [24]. Campbell’s work affirmed the significance of civic education and comprehensively introduced practices on classroom instruction, extracurricular activities, service learning, and the school’s ethos and summarized how these various aspects of schooling affect civic learning and engagement [25]. Fitzgerald et al. reviewed and compared the empirical research literature on the civic education pedagogy in the United States between 2009 and 2019, calling for more studies on K-12 civic education and evaluations of promising practices to increase youth civic participation and decrease political polarization [26]. Donbavand and Hoskins examined the causal claims between citizenship education programs and political engagement outcomes through a review of controlled trial studies [27]. Eidhof and de Ruyter focused on scholars’ different conceptualizations of civic and political self-efficacy and proposed a new framework for the study of self-efficacy in civic education research [28].

To sum up, among all the review work mentioned above, except for the two reviews in Chinese, they either focused on moral education or civic education, but did not address civic-moral education in an integrated way. Additionally, few reviews have been conducted to systematically reveal the pattern of Chinese civic-moral education research. This article aims to perform a scoping review to summarize the research on civic-moral education in China over the past 30 years, serving as a bridge to connect the vast body of work on Chinese civic-moral education with both Chinese and international educational researchers. This will not only provide international scholars with a convenient resource to understand the status quo of civic-moral education in China but also, through an overview, pinpoint new directions for further research. Moreover, a better understanding of the strengths of Chinese civic-moral education and the challenges it is facing may inspire educators to optimize their educational approaches.

Furthermore, due to the differences in political systems, economic development, and cultural values, civic-moral education in China differs from that in the Western countries. In terms of value goals, both China and the West emphasize the preservation of the public interest, but the Western countries emphasize the realization of individual social values through participation in democratic decision-making [29], whereas China emphasizes the realization of human social values through service to society. Although civic-moral education in China and Western countries presents a different pattern, the elements in the patterns are referable and transferable, which means that ideas and practices of civic-moral education in China may be useful for other countries from a new perspective. 

## 2. Methodology

Scoping reviews have many advantages, such as providing an overview of a field and being able to synthesize qualitative and quantitative research [30]. We used a scoping review to explore the existing literature on civic-moral education in China. Arksey and O’Malley offered the earliest description of a scoping review, together with a six-stage methodological framework [31], including (1) developing an appropriate research question, (2) identifying relevant studies, (3) selecting articles, (4) data charting and data extraction, (5) collating, summarizing, and reporting the results using thematic analysis, (6) and finally consulting with stakeholders to inform or validate study findings. Similar approaches were used to conduct this study.

We used “civic-moral education” or “social civic-moral education” (gongde jiaoyu or Shehui gongde jiaoyu) as keywords and searched the China National Knowledge Infrastructure (CNKI) and Wanfang Data. CNKI and Wanfang Data are the two largest and most comprehensive paper databases in China, which include almost all the papers published in simplified Chinese. The papers’ published years were set to be 1992–2022, and the search was completed by 12 December 2022. An initial database of 715 papers was created to include all the search results, of which 153 were from CNKI and 562 were from Wanfang Data. After four stages of screening and selection, 72 papers meeting the requirements remained. 

We have established inclusion and exclusion criteria to define boundaries for our systematic review [32]. The following inclusion criteria were used to complete the screening and selection process: (1) being Chinese articles; (2) being published in Chinese core journals; (3) full-text accessible; (4) being high-quality academic articles on civic-moral education. Exclusion criteria were also set up to make the screening procedure systematic: (1) no abstract or no access to the full text; (2) literature unrelated to civic-moral education, which means they only cursorily mention civic-moral education but not as a major subject; (3) non-academic articles, such as news reports, government documents, and political articles; (4) low-quality articles, referring to those articles with unclear arguments and poor argumentation process; (5) reviews. The literature screening and selection process is listed as follows and shown in Figure 1, with 72 papers finally included. Each step of the screening was stored in a separate file to maintain an audit trail.

Step 1: 90 duplicate records were removed. There are some articles in both CNKI and Wanfang Data, and in this step, we removed those duplicated records.

Step 2: 471 unrelated papers were excluded. By reading the titles and abstracts, we identified and removed articles merely containing the keywords, but addressing different topics.

Step 3: 8 non-academic articles were excluded. By reading the full texts, we identified and removed non-academic articles, such as news reports and official government documents. 

Step 4: 74 low-quality papers were excluded, with the remaining 72 left. By carefully reading all the relevant and academic articles, we identified and removed those articles with unclear arguments and poor argumentation processes.

All authors agreed on the screening methodology, with the leading author performing the main screening work and the second author separately screening the articles as well. Discrepancies were reviewed, discussed, and resolved by the author team to finalize the selection outcome.

After identifying the final 72 articles, we first categorized them into 10 empirical and 62 non-empirical articles and summarized the overall content (a complete list of the 72 articles can be found in the Appendix A, Appendix A). Then, we performed a thematic analysis. The leading and second authors reviewed all the articles independently and recorded potential themes and subthemes. Through a group discussion, we reached a consensus on the themes and subthemes. Three themes were finally identified from those selected articles: (1) the potential cultural resources on civic-moral education, (2) the approaches used in civic-moral education, and (3) existing challenges. Following this, we performed a thematic analysis of 72 papers on each theme to address the following questions: (1) What are the potential cultural resources in China that can help optimize civic-moral education? (2) How does civic-moral education in China work? (3) What are the challenges faced by civic-moral education in China? In order to fully answer these questions, all the authors reread all of the 72 papers, purposefully extracting from them the content related to the three themes. We then integrated the reading notes to sort out the contents of each theme and organized them in an appropriate way. 

## 3. Results

In this section, we first analyze the selected 72 articles by category to provide an overview. In the next three sub-sections, we present a literature review on each of the three themes. We summarize potential cultural resources on civic-moral education in the following three aspects: (1) traditional culture in the Axial Age, (2) ideas raised by modern thinkers, and (3) national leading policies on civic-morality construction. Then, we talk about the three main approaches used in civic-moral education: (1) building a civic-moral curriculum system, (2) transforming the social climate, and (3) making the most of the campus. Finally, we point out the existing challenges and potential solutions. 

### 3.1. Overall Condition and Content of the 72 Articles

Among the 72 selected papers, empirical studies are relatively lacking, and there are only 10 of them (see Table 1 for the categorization). Of the 10 empirical studies, 9 are social surveys. These surveys mostly form descriptive statistics of the current situation, and they rarely explore causal relationships or correlations, nor do they conduct follow-up surveys to analyze the trend of changes over time [33,34,35,36,37,38,39,40,41]. These articles were published in the period 2002–2021, showing civic morality at different points in time. However, comparisons across times cannot be made due to the differences in sample selection. The survey data show that Chinese people’s civic morality is good in general, although challenges remain. For example, people’s behaviors might be contrary to their inner morality [33,35]. The non-survey empirical study is an attribution analysis, which explores the main factors affecting secondary school students’ (non)civic-moral behavior [42]. This study found that students’ (non)civic-moral behaviors may be largely influenced by managerial deterrence and others’ elicitation, followed by environmental features and personal needs; students mainly consider self-interest in deciding whether to conduct civic-moral behaviors [42]. 

The 62 non-empirical articles can be broadly divided into the following five categories. Four of them specifically explore the conceptualization and significance of civic morality and civic-moral education in China. Twelve of them turn to the ancient famous Chinese educators and thinkers to explore the connotation of civic morality and their proposed methods of civic-moral education. Thirty of them present approaches to civic-moral education based on their personal working experience (especially in schools) and observations, among which three articles use public events (e.g., Olympics, epidemics) as teaching materials. Five of them discuss special issues on civic-moral education at the theoretical level, including the relationship between civic and private morality [43], the difference between Chinese and Western “civic”, and the reasons behind the difference [44]. Eleven of them point out some challenges of civic-moral education nowadays with explanations and potential solutions. The research themes and the corresponding number of publications are shown in Table 2.

### 3.2. Potential Cultural Resources on Civic-Moral Education

Scholars mainly studied three categories of cultural resources on civic-moral education in China: (1) traditional culture in the Axial Age, (2) ideas raised by modern thinkers, and (3) national leading policies on civic-morality construction. Traditional culture and the thoughts of the sages play a significant role in forming Chinese people’s moral values. In China, civic-moral education has a long history, starting from the Zhou Dynasty (1045–221 B.C.) and is highly emphasized. Confucianism is an important source of content for civic-moral education [13]; for example, their work, the Classic of Filial Piety (Xiaojing), describes specific norms of national civic morality [14]. Other influential ancient academic schools in civic-moral education, including Mohism and Taoism, also provide valuable resources for constructing civic morality. Mozi (the founder of Mohism)’s ethical ideas of “mutual love” (jianxiang’ai) and “mutual benefit” (jiaoxiangli) argue that loving each other can be mutually beneficial [45]. “Righteousness” (yi) and “virtue” (de), as the ideal status proposed by Lao-Tzu (around BC 571–471, the founder of Taoism), emphasizes the necessity of cultivating one’s own body and mind and assuming responsibilities equal to one’s abilities [46].

Despite rich resources in early history, the modern concepts of civic morality and civic-moral education were introduced into China around the early 20th century. Inspired by foreign theories and experience, advanced intellectuals involved in nation-building suggested modern civic-moral education. Many modern thinkers (Qichao Liang, Yutang Lin, Yuanpei Cai, Yat-sen Sun, Boling Zhang, etc.) raised the idea of advocating for new morality, cultivating new nationals, and providing a new ideological impetus for social change [47]. One of the most representative figures is Qichao Liang (1873–1929, a modern Chinese educator, political theorist, and social activist), who deplored the shortcomings of the weak civic consciousness of the people at that time. Similar to Qichao Liang, Yutang Lin (1895–1976) analyzed the reason for the lack of civic actions and argued that the traditional Chinese family lineage determined the lack of concern in public matters [48]. In 1902, Qichao Liang published an article entitled “On Civic Morality”, which advocated for civic morality education to be a reasonable way to transform society from traditional to modern. Qichao Liang maintained that civic morality reflects the ideal of cherishing the entire country as if it were one’s own family, so he believed that it is possible to inspire civic consciousness and civic morality with private morality. For example, it is easier to understand respecting seniors if people know filial piety towards parents in the family [49]. Yuanpei Cai (1868–1940) linked freedom, equality, and fraternity with the traditional Chinese morals of “righteousness” (yi), “forgiveness” (shu), and “benevolence” (ren) [50], promoting the integration of modern civic education and indigenous Chinese culture.

Traditional moral values are still deeply embedded in the minds of Chinese people today, regulating their behavior, and are regarded as important materials of civic-morality education. For a considerable amount of time after the establishment of the People’s Republic of China (1949), civic-moral education was ongoing without receiving high attention. However, with the development of the economy and the gradual withdrawal of public power from social life, problematic marketing behaviors, such as maximizing profit, have started to emerge [51]. In order to regulate people’s behaviors in the public domain and help people to further develop their civic morality, a particular landmark event took place in 2001: the government issued the Outline for the Implementation of the Construction of Civic Morality (later abbreviated as Outline 2001). This policy impacted civic-moral education in China, especially in schools, as it states that “schools are an important place for the construction of civic morality”, which has triggered many educators to devote themselves to finding ways of civic-moral education, and many practical experiences have been accumulated. This policy also highlighted the importance of traditional moral values in civic-moral education. In 2019, the Communist Party of China (CPC) Central Committee and the State Council issued the Outline for the Implementation of the Construction of Civic Morality in the New Era (Later abbreviated as Outline 2019). It is a revision of the 2001 version [52] and places new demands and more concrete guidelines on constructing civic morality, such as promoting polite and appropriate behaviors in cyberspace. Continuously, this new outline contains a summary of positive Chinese traditional values, also stressing the role of inheriting the traditional culture in realizing new moral demands. Many local governments at the provincial level have reproduced it on their official websites, encouraging civil servants and the public to study it carefully and use it to guide their actions, such as Zhejiang Province [53]. Led by the CPC Central Committee, all government departments, cities, industries, and units across the country have carried out active study and action. With the continuous efforts in constructing civic morality, the content of civic morality education in China has been gradually clarified, and the national leading policies have become new cultural resources on civic-moral education [52].

### 3.3. The Approaches Used in Civic-Moral Education

According to Outline 2019, civic-moral education needs both the heritage of former practices and innovation [54]. Scholars have offered some solutions to improve civic-moral education in China, including several aspects: reflecting on and utilizing traditional Chinese culture, building a civic-moral curriculum system, creating a social environment suitable for the development of civic morality, and making full use of a campus (such as building a hidden curriculum in schools). Actually, these approaches have been used to varying degrees in educational practices. However, since the implementation of each approach entails many operation regulations, scholars have offered their views on what can be optimized.

#### 3.3.1. Building a Civic-Moral Curriculum System

In China, there is a civic-moral curriculum system consisting of mandatory civic science subjects/courses in each school stage. For example, in high schools, the corresponding subject *Ideology and Politics* aims to improve the moral quality of students in four aspects: political identification, scientific spirit, awareness of the rule of law, and civic participation [55]. Starting from junior high school, students need to pass the civic science exams to graduate from each school stage.

Many specific ways to enable students to have better learning experiences and outcomes have been proposed in the literature. Weili Fu argues that civic-moral education can be carried out by educating about laws and law practices, where teachers not only clarify the relationship between rights and obligations but also extend it to the field of civic morality [56]. Wenliang Xiang focuses on age-specific teaching, insisting that the educational objectives of each age group should be created based on specific developmental needs in each age group and collect materials close to students’ lives for civic-moral education [57], including creating specialized textbooks and posters on bulletin boards. Setting role models [57,58], role-play [57,58], stories [59], and debates [60] are proposed to help students learn civic-moral values by putting themselves in the position of others through reflection and empathy and making moral judgments after better understanding the characters and the situation [51]. In addition, scholars propose that it is necessary for students to have hands-on practice, to achieve happiness through school or social service [34,54,61]. Particularly, through service and learning observations, students not only learn from experiences per se but also learn from their teachers. As school and class organizers, teachers are considered to be moral role models who are naturally followed and highly respected by students, and their behavioral and moral qualities have a profound impact on students [34,43]. 

#### 3.3.2. Transforming the Social Climate

Civic morality is a highly public subject, and civic-moral education is closely tied to the social climate [47,62], as individual preferences, perceptions, and attitudes are influenced by the social climate [63]. Creating a positive social climate can form a nurturing atmosphere of morality; specific measures include making full use of policies and laws, public opinion, an honor system, and practical activities [64].

Scholars have presented many concrete ways to create a positive social climate for civic-moral education. Firstly, publicizing rules, policies, and laws is conducive to the spread of fairness and justice [5,35,54,59], and involving all people in supervision [65] can strengthen moral restraint. Secondly, a qualified moral model with a good story [57,59,60] helps to shape the public opinion on civic morality [47]. Those moral models are sometimes called heroes as they sacrifice themselves for the public interest, especially during big public crises or events [54,66], such as the COVID-19 pandemic [67] and the Olympic Games [68]. An honor system that rewards people with outstanding civic-moral behavior may be powerful enough to encourage other people to follow [65,69]. Thirdly, as the internet increasingly influences people’s thinking, building a healthy internet culture is becoming more and more important [69]. Specific measures include strengthening online public opinion guidance [65,69,70,71] and implementing public opinion supervision [54,65]. Fourthly, the government can create more opportunities for social practice and encourage people to participate, since through practice, people can internalize moral norms into their own moral feelings and conduct [54]. Finally, the level of people’s civic morality is closely linked to the level of their knowledge, so it may be useful to promote education in general, including education regarding literature, art, and news [5,54].

#### 3.3.3. Making the Most of the Campus

School education occupies a large part of a person’s life [47], and it plays an important role in shaping students’ civic morality [5]. In school, the daily behaviors of students are supervised and corrected in time by teachers. In order to become qualified citizens, besides passive law-abiding aspects of civic morality, students are required to have a sense of virtue, the willingness to participate in social life, practical wisdom, and practical civic skills [35,44,72,73]. In terms of how to conduct civic-moral education in schools, many scholars have proposed concrete measures. Firstly, school media, such as campus radio and bulletin boards, are suggested to be important ways to disseminate relevant knowledge and values [34]. By reading the posters on the bulletin board or listening to the words on the radio, students learn more about civic morality. Secondly, an honor system may be valid to encourage students to participate in the practice of civic morality, in addition to transforming social climate [34]. Thirdly, schools can organize and guide students to participate in activities related to public welfare [54]. In many places in China, volunteering performance has already been taken into account when evaluating students [56]. Finally, allowing more students to participate in the management of class and school affairs may enable students to continuously develop the ability of civic participation and thus care more about public affairs and civic morality in the whole society [6]. 

### 3.4. Existing Challenges and Potential Solutions

There are two major challenges with civic-moral education in China: the emphasis on intelligence over morality and the emphasis on knowledge over action. Ling Yang and Xiuhui Zhou presented, that in China, parents pay much attention to students’ intellectual development, and some parents even take it as the only indicator to measure their children’s achievement [36,74]. Xing Zhang argued that Chinese universities pay less attention to students’ civic morality compared to test scores, thus failing to create an atmosphere that values civic morality [74]. In addition, Wenliang Xiang and Xing Zhang realized that in some schools, civic-moral education is dominated by the classroom teaching of bookish knowledge without incorporating practice; while it is undeniably important to teach related theories, practice is an important and essential way for students to learn how to engage in public affairs [57,74]. Licheng Qian and his colleagues found that contrary to expectations, the exam alone seems to not be able to facilitate students’ learning of social ethics [75]. For these challenges, scholars’ proposed solutions are in alignment with the approaches suggested above. For example, many scholars have proposed to make practice one of the key elements of civic-moral education [34,54,61]. If the aforementioned approaches can be better implemented, the situation of belittling morality and practices might also be improved. 

## 4. Discussion

This paper is a scoping review that systematically summarizes the pattern of Chinese civic-moral education research in three decades. It not only offers scholars a quick overview of Chinese civic-moral education research, but also adds to the general literature on civic-moral education, which may inform educational practices beyond China. To more succinctly show our main points, we first summarize the salient features of civic-moral education in China so that readers can understand the current status of research in this field at a deeper level. Secondly, based on a comprehensive overview of the Chinese literature on civic-moral education, we point out three limitations in those previous studies, which provide viable ideas for future research in this field. Finally, we reflect on our own research work and summarize several limitations.

### 4.1. Characteristics of Civic-Moral Education in China

This scoping review searched the two largest academic databases in China and collected 715 relevant papers on the topic to summarize the research achievements in civic-moral education in China in the past 30 years. Through screening, 72 articles were selected, most of which are non-empirical and make arguments based on personal experiences and theoretical discussions. Only 10 are empirical studies, which focus mostly on the description of the current status.

By summarizing the papers on civic-moral education research, we have depicted the characteristics of civic-morality education in China, which may facilitate international researchers and educators to better understand current practices in China. We found that civic-moral education is largely supported by rich Chinese cultural resources [13,14,15,16,45,46,65], which is also in line with Cheng et al.’s review [21]. Traditional culture and the thoughts of famous educators play a uniquely significant role in forming Chinese people’s moral values. With a long history and all-around layout of traditional culture, China employs traditional culture as an important vehicle for civic-moral education. Many articles start from the important thoughts on Chinese traditional culture and analyze their values to today’s civic education [13,14,45,46]. For example, the concept of “righteousness” (yi) [47] and the culture of “ritual” (li) [65] of Confucianism are conducive to social harmony. Although the modern concept of civic morality was derived from Western countries, Chinese scholars have incorporated ideas from traditional Chinese culture into modern civic-moral education. 

By and large, civic-moral education in China is top-down and strongly government-driven. In addition to Outline 2001 and Outline 2019, mentioned by scholars as contemporary cultural resources, there are also national-level value-shaping actions that similarly succeeded much from traditional moral thoughts. “Eight-honor and eight-shame” and Core Socialist Values are typical examples. In March 2006, “eight-honor and eight-shame” was proposed, which specifies the behaviors to be advocated [76], including honor to those who love the motherland, shame to those who harm the motherland, etc. “Eight-honor and eight-shame” absorbs the traditional Chinese concept of honor and shame, as it honors loyalty, benevolence, righteousness, honesty, tolerance, etc., and shames betrayal, selfishness, evil, lies, etc. [77] With the top-down promotion, “eight-honor and eight-shame” has taken root in people’s hearts [78]. In November 2012, Core Socialist Values was announced, including three levels of values: national, social, and personal [79]. Driven by the CPC Central Committee, Core Socialist Values have become the important content of civic-moral education and the guide for citizens to cultivate their own civic morality [80]. It carries over elements from traditional moral thoughts as well, such as “fairness” adopted from “righteousness” (yi) [13] and “harmony” adopted from “universal love and non-aggression” (jian’ai feigong) [45]. 

Additionally, we found that civic-moral education in China is systematic and comprehensive through a series of approaches. The formal educational arrangements, such as a systematic curriculum covering all school stages, textbooks, and examinations, officially guided and supported, play an important role in civic-moral education. Simultaneously, extracurricular activities, campus culture, and social climate are created to provide a positive condition for civic-moral education. A blizzard of literature illustrated the action of the government to influence social climate through policy [39], mass media [66], activities [67], etc. The school encourages students to participate in social activities and cultivate the spirit of civic morality in practice. To sum up, scholars have started to pay more attention to the implicit educational system and propose concrete approaches aiming for a synergistic effect of various components of the civic-moral educational system. 

### 4.2. Directions for Future Civic-Moral Education Research

Previous research in the last 30 years on Chinese civic-moral education focuses on related cultural resources, approaches, and challenges. Even though meaningful findings have been discovered, there are some shortcomings in these studies, thus showing directions for future research. Firstly, most of the theoretical literature attaches great importance to discussing abstract values rather than exploring how to conduct civic-moral education based on students’ developmental needs. Future literature can focus on the interconnectedness of civic-moral education and human development and use theories of developmental psychology to better discuss how to optimize educational approaches and facilitate students’ development. Future studies may also analyze the differences across school stages in terms of civic-moral education and how these practices cater to students’ age and maturing level in each stage.

Secondly, the conceptualization of civic-moral education needs further work. The relationship between the protection of individual interests and civic morality has seldom been explored in existing studies. These two are not contradictory to each other. However, previous research often presented Chinese civic-moral education in a collectivist narrative [66,68], without much attention to individuals. How individual interests are related to collective interests needs further elucidation in future studies.

Thirdly, empirical studies are relatively lacking. Some studies pointed out the challenges of civic-moral education based only on personal experiences or observations [70]. Some studies proposed solutions or particular approaches to solve these challenges without empirical evidence of their effectiveness [15,63]. Descriptive statistics based on simple surveys cannot provide information regarding the reasons behind certain phenomena [41]. More empirical studies using relatively advanced quantitative and qualitative methods are needed to test existing viewpoints and proposed approaches. An interventional study can serve as a good example: Yao et al. found that *Dizi Gui* instruction, a Confucian classical approach, can enhance Chinese adolescents’ peer relationships and teacher-student relationships through the mediating effect of prosocial behavior [81]. The factors affecting the effectiveness of civic-moral education through an in-depth analysis are lacking as well. Future research may apply empirical investigations to test the effectiveness of certain civic-moral education practices and the factors affecting their effectiveness.

In conclusion, some limitations remain in the existing literature, and future research on civic-moral education in China could focus more on these directions: (1) investigations of how to conduct civic-moral education from the perspective of human development, (2) valid discussions on the relationship between individual interests’ protection and civic morality promotion, (3) more empirical evidence for many arguments and analyses at the theoretical level, (4) more empirical evidence to prove the effectiveness of educational methods for civic morality and the factors affecting the effectiveness of civic-moral education.

### 4.3. Limitations

In this study, we have identified and summarized the cultural resources, approaches, and challenges in Chinese civic-moral education mentioned in the selected articles, providing a quick overview of Chinese civic-moral education research and a reference for scholars to conduct future studies. However, there are also limitations in this study. Firstly, literature written in English on Chinese civic-moral education is not included. Secondly, we did not specifically review the literature on civic-moral education according to school stages because many scholars treated civic-moral education as a whole when discussing the values of civic-moral education. Future studies may focus on a specific school stage to better understand the characteristics of civic-moral education in China, as well as relevant research.

## 5. Conclusions

After analyzing and summarizing the selected papers in this scoping review, we divided our findings into three parts. First of all, the cultural resources of civic-moral education are prominently displayed in the thinkers of the Axial Age, who put forward many core ideas that have played a very important role in shaping Chinese moral concepts. Nowadays, China has a well-established system of cultivating civic morality, especially in schools. Secondly, the approaches to civic-moral education in China include building a curriculum system, transforming the social climate, and fully utilizing the campus environment. Finally, the emphasis on intelligence over morality and the emphasis on knowledge over action are the remaining challenges in civic-moral education in China. Future research can be conducted in the following areas to further advance the field and inform practices: (1) the relationship between civic-moral education and human development, (2) empirical research with advanced quantitative and qualitative methods to better understand challenges in civic-moral education and show the effectiveness of certain approaches, (3) how to protect the interests of individuals while serving the public interests when facing concrete civic morality issues.

## Figures and Tables

**Figure 1 behavsci-13-00819-f001:**
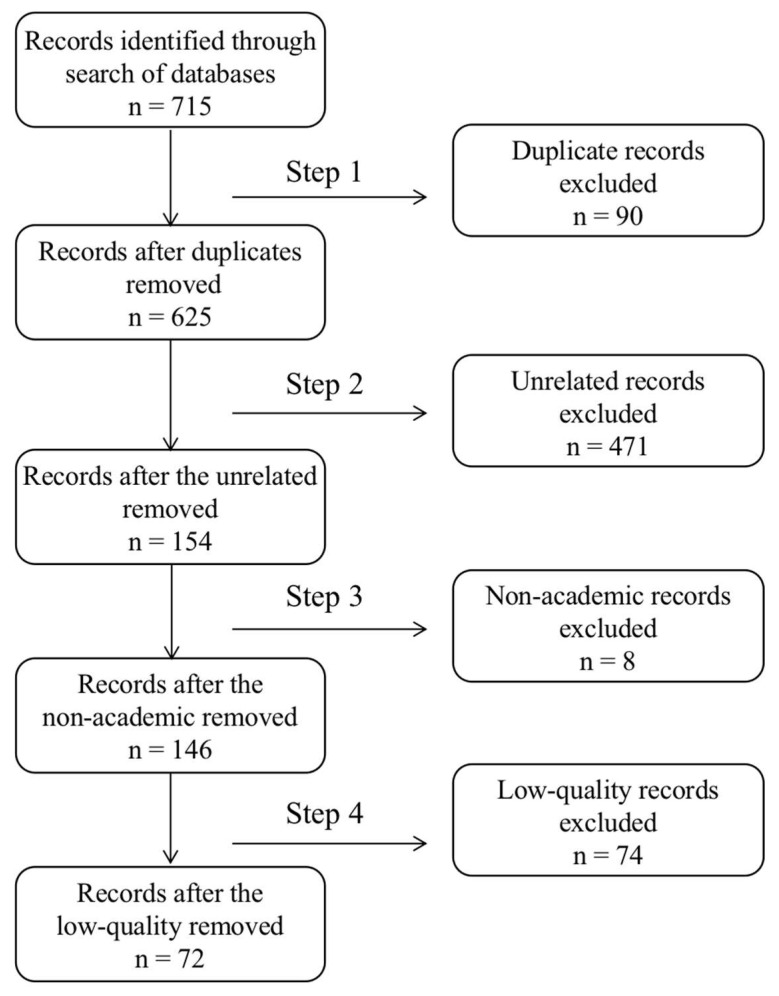
Flow diagram of the article selection process.

**Table 1 behavsci-13-00819-t001:** Classification of the 10 empirical articles.

Category	Number	Typical Articles
Social surveys	9	Survey and analysis of civic morality of college students in the new era [33]
Attribution analysis	1	Elements of influence on students’ civic-moral behavior and their motivational basis [42]

**Table 2 behavsci-13-00819-t002:** Classification of the 62 non-empirical articles.

Category	Number	Typical Articles
Conceptualization and significance	4	The classification framework of civic-moral education content: An analysis of the dilemma of public moral education in China [2]
Ideas from famous educators	12	Confucius’ ethical thought of “righteousness” and the practical path of civic-moral education [13]
Approaches	30	Significance, orientation and path: Social morality education adapting to the governance of unexpected public crise [8]
Theoretical discussions on special issues	5	On the promotion of Gongde in Chinese world from liberalism vs. communitarianism debates [7]
Challenges and potential solutions	11	The dilemma of public morality in contemporary China and its enlightenment to the school public morality education [6]

## Data Availability

The Appendix A includes data that were used throughout the stages of the analysis.

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
