# Peer review of "Civic-Moral Education Research in China (1992–2022): A Scoping Review"

_behavsci, 2023, doi:10.3390/bs13100819_

Round 1

Reviewer 1 Report

Dear authors, 

I have some recommendations for you. Please consider it. 

1. Introduction

The scope of the literature should be expanded. The introduction includes very few relevant studies. You should explain why the research is necessary and important clearly. You can use these studies for your introduction and methodology sections. 

https://www.atlantis-press.com/proceedings/ichess-21/125967394

https://www.tandfonline.com/doi/abs/10.1080/00313831.2021.2021439

2. Methodology

On page 2, it should be explained in more detail how the studies were selected. What are the selection criteria? Mention, please.

On page 3, how the thematic analysis is conducted should be explained more clearly.

3. Results

How the findings are analyzed is sufficient to provide a general framework for civic-moral education in China.

4. Discussion

It is good to point out the gaps in the literature to provide a perspective for future research.

Considering the main reason for the importance of this research, based on the lack of literature, will make the study stronger.

I want to see again your manuscript after revisions. 

Fair quality

Reviewer 2 Report

This is a very interesting an important review article about civic moral education in China. I think it is generally very well written and does indeed make a good and interesting novel contribution. 

Here are my suggestions:

a) Abstract: The abstract should include 1-2 sentences to make the core finding of the review more clear. 

b) Introduction: At the end you could add please why your review is needed and add (if existing) similar previous reviews. 

c) The methods section described really well how the papers which were included were selected. However, i am not sure how/why the 3 themes were extracted. Which qualitative procedure was followed here? More details would be needed i think.

d) The results section needs to make more clear how the content (e.g., Building a civic curriculum system) is related to the content of the papers which were selected in the review. Is the results a summary of what was suggested in those papers? How does the results relate to the papers that were included in this review?

Round 2

Reviewer 1 Report

Dear authors, you revised your manuscript partially, but it needs more revisions. Please consider my all notes of previous evaluations. You mentioned adding some studies to the manuscript, but I could not see them in the References. Please highlight the all revisions in the text. 

 Minor editing of English language required.

Author Response

Thank you very much for your advice. Attached are the instructions for the changes.

Reviewer 2 Report

Thank you. All my issues have been resolved in the revision now.

Author Response

Thank you so much. Wish you all the best!

Round 3

Reviewer 1 Report

Thanks for your efforts and clarafications. Good work, congratulations.

Minor editing of English language required.